# Exploring the Sources of Stress and Coping Strategies of Turkish Preschool Teachers

**DOI:** 10.3390/bs14010059

**Published:** 2024-01-16

**Authors:** Sinan Koçyiğit, Türker Sezer

**Affiliations:** 1Department of Preschool Education, Kazım Karabekir Education Faculty, Atatürk University, Erzurum 25240, Türkiye; kocyigit@atauni.edu.tr; 2Department of Preschool Education, Education Faculty, Bolu Abant İzzet Baysal University, Bolu 14030, Türkiye

**Keywords:** stress, job stress, preschool teachers’ stress, preschool education

## Abstract

This paper identifies preschool teachers’ sources of stress, the times they experience high stress, and their ways of coping with stress levels. The study was conducted using a phenomenological design, one of the qualitative research models. The data were collected through semi-structured interviews with 36 preschool teachers working with children aged 0–6 in state and private schools. As a result of the content analysis, stressors were identified primarily at both interpersonal (positive and effective interactions) and organizational levels (school management and leadership style). It was also found that intense stress was experienced when children were difficult to control, such as during sleeping, eating, and cleaning. Finally, these results confirmed that the teachers used active/active behavioral, and passive/avoidant coping strategies. However, it was understood that preschool teachers perceived stress negatively, and did not see stress as a personal development situation. These results are discussed in terms of their ramifications for preschool education.

## 1. Introduction

Stress is an emotional state that can be experienced by individuals in almost any occupation. It is a universal emotion that arises when the limits of the body are exceeded. Stress manifests itself through a broad range of physical and mental symptoms, resulting from maladaptive conditions that push the boundaries of the individual [1,2,3]. From a physiological perspective, experimental evidence supports the notion that stress amplifies susceptibility to coronary heart disease and stroke in adulthood [4]. In addition, studies by Lupien et al. [5] demonstrated the impact of stress on brain structures involved in cognition and mental health, both early and later in life. Furthermore, subsequent research by Lupien et al. [5,6,7] consistently showed that stress induces memory deficits and structural differentiation in key neuroanatomical regions: hippocampus, amygdala, and frontal cortex. In terms of psychology, research on women has shown that stress is significantly associated with more frequent thoughts of suicide [8]. It disrupts the harmony of the individual for a short or long period [1,3]. Olpin and Hesson [9] propose that stress encompasses three key aspects. Firstly, it involves the individual’s subjective perception of stressors, which can have both positive and negative effects on their life. Secondly, the experience of stress depends on how individuals react to life events rather than the events themselves. Finally, stress represents a challenge to the body’s capacity to adapt, where a strong and healthy coping ability leads to positive outcomes, while a weak ability results in negative outcomes. In conclusion, stress is a multifaceted construct that exerts an influence on employees’ performance outcomes [10].

Job stress is defined as experiencing negative emotions related to working [11]. It is often associated with emotional exhaustion and burnout [12]. Teacher stress, a growing concern worldwide and a type of job stress, can be defined as conditions that negatively impact physiological, psychological, and social well-being in the school/classroom [13,14,15]. The theoretical framework for the study of teacher stress in this study was the transactional stress model [16]. The model suggests that the components identified as environmental stressors, individual perceptions, organizational roles, and individual characteristics should be treated as having a multidirectional causal relationship. It has also been suggested that these components impact individuals’ short, medium, and long-term responses [16,17,18]. Stress is considered to be one of the most important variables of educational institutions, and it affects teachers [19], and it is fed by different sources such as the physical environment, the children and their parents, and the administration. Job stress arises from the detrimental impact of the work environment on an individual’s well-being and resources [20]. Individuals experience varying levels of stress, characterized by the expression of negative emotions such as frustration, anxiety, work anxiety, impaired emotional regulation, fatigue, and sleep disorders [21,22,23]. However, another variable associated with stress is “emotional labor”. Emotional labor is defined by Hochschild [24] as “the management of emotions to create a publicly observable face and body image” (p. 7). Recent research has produced evidence that reveals the relationship between emotional labor and job stress [25,26]. Consequently, it is understood that stress has multifaceted effects on teachers. Contemporary research presents substantial empirical evidence underscoring the impact of stress on teachers, as well as the broader ramifications on students and school outcomes [27,28,29,30,31].

Stating that stress should not always be perceived as a bad situation, researchers emphasized the level of stress and stated that low levels of stress could motivate teachers. In contrast, high stress levels can reveal negativities [32]. Teachers who experience high levels of stress may experience increased burnout and decreased job performance [33,34]. In teaching, which is one of the stressful professions [35], stress can be exacerbated by the limited time available to fulfill additional tasks other than teaching, such as planning, preparation, accountability for average student performance, supervisory roles, extracurricular activities, and monitoring [36]. At the same time, demands on the teaching profession, lack of resources, difficulties with financial planning, burdens of responsibility, challenging behaviors of children, characteristics of parents, and curriculum expectations are factors that contribute to stress in the profession [11,19,37,38,39,40]. Moreover, it is emphasized that students’ learning, discipline problems [29,41,42,43], time pressure, low income level, low social status, excessive workload, trends or developments in education, and problems related to school management also have an effect [44,45,46].

Managing stress is important for preschool teachers so that their traumas do not interfere with achieving their academic goals and having a better quality of life personally and professionally [47,48]. Research supports this theoretical perspective, and suggests that being a preschool and kindergarten teacher, one of the teaching professions, can be stressful, especially when working with children of low-income families, children at risk, and/or children with behavioral problems [13,15]. As preschool teachers’ stress increases due to children’s behavioral problems, the emotional climate in the classroom may be negatively affected. For example, teachers assigned to classrooms with more children with behavioral problems may experience more stress and may not be able to manage the group effectively in the absence of collaboration [49]. Research into early childhood education suggests that teachers play a critical role in creating and sustaining high-quality learning environments [50]. These environments are important for children’s cognitive, language, and social-emotional development [51,52,53,54]. However, it can be assumed that the ability of teachers to provide a high-quality learning environment may be adversely affected by the stress they are experiencing [49].

Several studies have been carried out in Türkiye to investigate the stress experienced by teachers in general. These studies have been conducted with teachers at the elementary, middle, and high school levels and have been mostly quantitative in design. They examined variables such as sources of stress [44,55], organizational stress [19], stress and job satisfaction [56,57], stress and self-efficacy [58], stress and self-esteem [59], and stress as occupational health [60]. However, few studies have been conducted regarding the stress experienced by preschool teachers in Türkiye. These studies investigated variables such as management process problems [61], stress symptoms, sources, organizational stress [62], stress and classroom climate [30], and professional burnout [63]. These studies are quantitative in nature. Similarly, the international studies reflect the results of the research in Türkiye, as described above [13,15,41,43,64,65,66]. 

Preschool teachers encounter stress as a result of their engagement with children in the younger age group [59]. Teachers experiencing emotional burnout are less likely to interact with children, which can lead to lower-quality interactions with children [27]. Teachers’ psychological well-being impacts children’s development and the nurturing and learning classroom climate in early care and education [67]. Therefore, a better understanding of the sources of stress for preschool teachers is critical to creating a classroom climate that will reduce stress and ultimately support young children’s development. In this regard, research based on examining different variables and using qualitative methods is considered necessary [41]. It is with this in mind that this study aims to explore how preschool teachers perceive and cope with stress and sheds light on their unique experiences.

### Preschool Education and Preschool Teacher Education in Türkiye

Preschool education is not compulsory, and it varies according to the age group and the type of institution. There are independent kindergartens (36 to 69 months), kindergartens in primary schools (57 to 69 months), and nursery schools (3 to 36 months) in a variety of educational institutions. There are also private children’s clubs (mixed age) and religious kindergartens (36–68 months). The Turkish Preschool Education Program was updated in 2013. Its goals are to support all areas of children’s development, help students acquire good habits, prepare them for primary school, provide a common educational environment for children from different backgrounds, and support their Turkish language development. The program is child-centered. It is flexible, eclectic, balanced, play-based, and dynamic. It also promotes creativity and learning by providing a stimulating environment appropriate to children (learning centers, etc.). Family involvement, education, and guidance are emphasized in the program. In addition, the program asks teachers to use a multidimensional approach to assessment (e.g., portfolio). Finally, the program encourages teachers to organize large-group (whole class), small-group (collaboration), and individual (individual differences) activities in a balanced manner, plan activities in an integrated manner, and implement activities adapted for children with disabilities (if any) [68].

Universities carry out initial teacher training programs for preschool education in Türkiye. Today, the faculties of education are the main source of teacher training. All of the primary level teacher education programs last for four years. The Teacher Education Program for primary level was revised by the Council of Higher Education in 2018. Key factors that take the center stage include the Bologna Process, studies on quality and accreditation, the imperative of developing core curricula in Türkiye, the escalating importance of ethical, moral, spiritual, and cultural concerns, and the goal of fostering parity in international student exchange programs. The core curriculum has been renewed and the number of elective courses has been increased to create a framework of qualifications in line with the European Qualifications Framework for Higher Education [69,70]. 

## 2. Present Study

Identifying modifiable variables that contribute to the stress experienced by preschool teachers, who represent a distinct group of educators, holds significant importance [43]. By identifying these variables, we can gain a deeper understanding of teachers’ present experiences within the classroom, thereby providing an opportunity to examine and analyze their unique dynamics. Furthermore, this study responds to the call for systematic research that includes different variables, including the differentiation between preschool teachers working in private and public/state schools, and the age range of the children they interact with, from infancy (0–36 months) to early childhood (36–72 months). This broader scope goes beyond previous research, which has mainly focused on teacher and child behavior, and contributes to a more comprehensive understanding of the preschool classroom environment [41]. Therefore, gaining insight into the underlying factors that contribute to the stress experienced by teachers working with different age groups in different institutional settings may facilitate the development of more effective coping strategies. This, in turn, has the potential to enhance the overall well-being and resilience of these teachers, ultimately benefiting both their professional practice and the educational experiences of the children in their care. Another reason for this study is that the studies conducted with preschool teachers in Türkiye are organized in accordance with the quantitative method and have simple linear relationship findings based on relatively few cases. 

Theoretically, the stressors in the preschool teachers’ environment, the teachers’ individual characteristics, and their reactions to stress have been assessed [18]. Thus, in terms of organizational structure, the status of schools as private or public, nursery or kindergarten, and in terms of individual characteristics, the experience of teachers (working with younger and older children) were taken into account [16,17]. In addition, this study examines the hypotheses that teachers externalize the source of stress [71,72] and that they do not use active cognitive strategies to cope with stress [73]. For these reasons, this study of preschool teachers’ sources of stress and ways to cope with stress is considered important because it reveals preschool teachers’ views holistically. The holistic structure expressed here refers to considering examples of institutional differences in preschool education in Türkiye (prekindergarten, kindergarten, and nursery school), examples of age group differences among children (0–3 years old, 3–4 years old, and 4–6 years old), and public and private preschool education institutions.

## 3. Method

This research focuses on the phenomenon of “teacher stressors”. A qualitative approach called transcendental phenomenological design was chosen to gather data on preschool teachers’ stress experiences. The focus was on the description of the experiences of the teachers in relation to stress [74] (p. 34). This design is suitable for studying intense, emotional human experiences [75]. 

By using semi-structured interviews as the primary data collection method, the study aimed to uncover the experiences, perceptions, and meanings attributed by preschool teachers to the phenomenon. The phenomenological design aligns with the study’s purpose and process, as it aims to understand how participants experience the phenomenon [76] (p. 408). The choice of interviews aimed to capture the essence and meaning of the experience, making it a preferred method for such studies [75] (p. 25).

### 3.1. Participants

The participants of this study, which was reached through a purposive sampling method, were 36 preschool teachers working in the province of Erzurum, which is located in Eastern Turkey. In selecting participants, the age group of the children in the classroom and the criteria of working in a public or private preschool education institution were considered (see Appendix A). The respondents were all female teachers, and their participation was on a voluntary basis (see Appendix A). In fact, the profession of preschool teaching is a profession chosen mostly by women. The proportion of female teachers in Türkiye is approximately 94% (n = 121,786), while the proportion of male teachers is approximately 6% (n = 7667), according to education statistics on the distribution of preschool teachers by gender. This ratio was similar in the province of Erzurum, where the participants of the study were identified (92.5% female teachers, n = 587, 7.5% male teachers, n = 48) [77]. Therefore, a few male teachers were also contacted, but these teachers did not voluntarily agree to participate in the research. The information regarding those participating is shown in Appendix A (see Appendix A). 

According to Appendix A, 50% of the participants work in private schools and 50% in public schools; 16.7% work with children aged 0–3, 33.3% with children aged 3–4, and 50% with children aged 5–6; 16.7% of the teachers work in nursery schools, 66.7% in kindergartens, and 16.7% in preschools; 27.8% have 1–3 years of seniority, 38.9% have 4–7 years of seniority, 19.4% have 8–11 years of seniority, and 13.9% have 12–15 years of seniority; and all the participants are female. 

### 3.2. Data Collection Instruments

In data collection, the “Individual Information Form” was used to determine the demographic characteristics of the participants (gender, seniority, school type, age of the child group, etc.). In order to identify sources of stress, periods of stress, and strategies of coping with stress, the semi-structured interview form developed by the researchers included questions like “What/which situations do you consider to be a source of stress while teaching?”, “In what situations do you experience stress as an educator?”, and “How do you deal with sources of stress?”. Interviews with the participants were conducted by the first author. The first author conducted the interviews with the participants. This researcher has experience in and published articles on qualitative research methodology. The first draft of the form was thoroughly reviewed by experts, and its final structure was refined through feedback from five preschool teachers to improve the clarity of the questions. Appointments were made with 36 preschool teachers before the inter-views. The purpose and process of the research were explained to the participants, and the interviews were conducted face-to-face at the school. One interview lasted approximately 30 min. In order to prevent possible data loss during the interviews with the participants, it was stated that the answers to the questions in the interview form would only be listened to by the researchers and after approval; the interviews were recorded with a voice recorder and were listened to by the participants, and their consent was obtained. The recorded data were transferred to a computer in audio files, listened to by the researchers, and written in Word files.

### 3.3. Data Analysis 

Content analysis, one of the qualitative data analysis techniques, was used to analyze the data. First, a symbol and number were assigned to each participant to make the data anonymous; the data were transferred to a Word file. The researchers carried out a thorough reading of the data, analyzing it line by line (word by word) using an inductive approach. Inductive content analysis was used to identify concepts and relationships that could effectively capture the collected data. The data were first coded to identify similarities and differences between them. Subsequently, the large data set was broken down into smaller data units for further analysis [77,78]. Following the coding process, a reliability analysis was conducted to ensure the robustness of the findings. The reliability of the study, calculated using the Miles and Huberman formula [reliability = 49/(49 + 5)], was determined to be 90% [79] (p. 64). 

The researchers identified commonalities among the codes developed, which led to the establishment of categories and themes that form the basis of the findings section of this study. To ensure the validity of the themes, a careful review was conducted to assess their consistency with the theoretical and conceptual framework [16,17,18]. This process facilitated the refinement and finalization of the identified themes and strengthened the overall coherence and credibility of the study findings. 

When quoting direct quotations from the interviews, these symbols and numbers were used. An inductive approach was used in the analysis process, and similar codes were grouped under certain concepts and themes. The coding used to anonymize the data is as follows.

P: Private

S: State

T: Teacher

N: Nursery (for children ages 0 to 3)

Pr.: Preschool (for children ages 3 to 6)

K: Kindergarten (for children ages 4 to 6)

### 3.4. Checking Research Quality

To check the quality of our research, we used the COREQ (Consolidated Criteria for the Reporting of Qualitative Research) checklist [80]. We considered the checklist’s 32 items and reviewed how well our research met each item. As a result of our review, we found that our research met the criteria. In this context, we can state that our research took into account qualitative research procedures (see Appendix A).

## 4. Results 

The presentation of the findings of the study takes into account the interview questions that were used in the data collection process, the concepts and categories that emerged from the responses to these questions, and direct quotations stating the opinions of the participants, supporting the concepts and categories.

### 4.1. The Sources of Stress

In the interviews, the teachers were asked the following question: “What/what situations do you consider to be stressful when teaching?”. Analyzing the responses to this question produced the results shown in Figure 1.

The analysis resulted in three stressors: parents, children, and institutional/management. Within the initial category, all participants expressed that their encounters with ‘parents’ constituted the most significant stressor in their work lives. The “parents” category was coded as follows: communication problems with family, inappropriate expectations of family, negative attitudes toward preschool education, disobedience of school rules, and use of family/teacher communication tools. Examples of quotes from the respondents for these codes are shown below:

Communication weakness: “*…that’s why we usually have problems communicating with the parents. Because their mindset, and approach is that if I give you money, you must do everything perfectly. We get stressed because they project this duty onto us. We are afraid after the child experiences a negative situation. Our stress is about how the parents will react, how it will be, will this parent complain to our principal, will there be tensions.*” (PTN-1).

Inappropriate expectations: “*…the family asking me questions about the children’s nutrition. They come to me with nurturing questions (like a nurse) like give them so much of this food, they like this, don’t give them that… This situation has made me feel very sad. First, we support the self-care skills of the younger age group (3 years old), then we support other skills. But to focus only on nutrition is something that is difficult for a preschool teacher.*” (STPr-3).

Negative attitude: “*Parents’ behavior and attitudes are negative. They have high expectations of the school. They have expectations that are above the level of their own children. They want everything they say to be done.*” (PTN-5).

Resistance to obey school and classroom rules: “*…behavior of the family against the rules of the school. After bringing the children to school, they want to stay with the child.*” (STPr-6).

Communication tools: “*WhatsApp groups can make our days’ tense. These kinds of situations can be a source of stress for us. Also, we cannot convince the parents in a situation related to children. Poor cooperation and miscommunication between school and family make us feel stressed. Even after our return home, these problems are still on our minds.*” (STK-5).

The second category, encompassing children, was delineated through various codes, including the presence of younger-aged children, children exhibiting behavioral challenges, those who had not yet developed self-care abilities, children requiring special education, instances of infectious illnesses, attendance issues, and a wide age range of children within the classroom setting. Examples of attendee quotes for each of these codes are provided below:

Child age: “*I mean, of course, little kids are a lot of stress for me. In particular, I have been working with 4-year-olds. The 5-year-olds are suitable for preschool, but the 4-year-olds are a bit more difficult. When I’m warning them and so on, it’s a little harder to control them because they’re very young. I have a hard time in that regard*.” (PTPr-2).

Children’s behavior problems: “*…the children are hitting each other, falling down and hurting each other. It is very stressful for me*.” (PTPr-1).

Self-care skills of the children: “*…toilet training and self-care skills stress me out. Especially feeding times of young children stress me out. Because parents have very high expectations in this area. I find it difficult to cope with statements such as “let him/her eat everything”, “let me prepare the food I want”, “let my child eat what he likes”, “my child is very special”.*” (STPr-4).

Inclusion–mainstreaming: “*…here are the inclusion children for example. Typically developing children can have the education you provide, but children with special needs cannot, and the whole class may be in disarray. But this is our duty, we have to*.” (PTPr-3).

Illness: “*…sending children to school during the infectious disease phase.*” (STPr-6).

Attendance: “*…the lack of continuity of children’s attendance at school.*” *(STK-4).*

*Age ranges of children:* “*…the high age range of the students enrolled. In other words, there are children of different age groups in the class. This creates chaos and causes stress. Classroom management is not effective*.” (STK-6).

The concluding category, pertaining to institutional/management factors, was designated through the following codes: substandard physical infrastructure of the school, lack of support staff, excessive child size, demanding full-time work responsibilities, inappropriate practices during orientation week, arrangements for special occasions (national and religious observances), organizing field trips, and conflicts arising between teachers and administrators. Examples of the participants’ views on these codes are presented below:

Physical conditions: “*…We cannot buy the materials we want to use because of their prices, and there are physical inadequacies in the institution*.” (STK-3).

Support staff: “*…I have to take care of things that are not in my job description because there is no assistant in the classroom. I neglected the classroom at that time. Do I take care of the classroom, do I go to the bathroom, and which one do I follow? Especially in the first days of school, I go through a very problematic process until order is created.*” (STPr-4).

The number of children: “*…l has many children in my class, class size over 30 is really difficult…*” (STPr-6).

Working hours: “*It is exhausting because it is an entire day. It is very exhausting because we are very tired from morning to night. We are also experiencing the stress of it all.*” (PTN-1).

Adaptation program: “*…in my professional life so far, the only thing that is a problem for me is the crying of the new students. I am afraid they will get used to it and stop crying. But we are slowly getting over it.*” (PTN-6).

Special days: “*We get stressed out preparing for special days. For example, the 23rd of April (National Independence Day). … Getting the children to work is stressful for them and for us. Because you have to do a show, you have to do a presentation, and you have to put a lot of effort into it. So, both the child and the teacher can be stressed out in the process of doing that.*” (STPr-2).

Field trips: “*…the permissions for the field trips and the worries about keeping the children under control during the field trip are a source of stress to me.*” (PTPr-5).

Management: “*…the administration’s view of preschool education. Contrasting the teachers’ and administrators’ views on educating*.” (PTPr-6).

### 4.2. Teachers’ Ways of Coping with Stress

When interviewing the teachers, the following question was asked: “How do you cope with stressful situations?”. The analysis of the answers to this question produced the results shown in Figure 2.

The preschool teachers reported that they used ways of coping with stress in the following categories: Active behavioral and passive/avoiding. Teachers cope with stressful situations they experience with parents, which is the most intense stressor, using an active behavioral and passive/avoidant strategy. Examples of opinions related to these codes are presented below:

Effective communication: “*…I am always in touch with the parents and explain every situation to them. I try to explain to them: The more you cooperate with me, the more progress we can make*.” (PTN-1).

Positive approach: “*…I think leaving the house with positive energy is necessary. Let me explain I have a positive attitude toward my children, and they give me positive energy. In this way, I try to stay away from situations that are stressful for me.*” (PTN-4).

Sharing: “*…When the parents show negative behavior, I invite them to the school and ask them to observe us for a whole day. I ask them to take part in the events and activities that we have. By doing so, I ensure that I share all situations with them…*” (STK-2).

Family involvement: “*I usually go to the parents’ homes to ensure their participation. That’s why I do it because they don’t visit the school. They participate by saying “OK, teacher, we are going to do it”, “we are going to do it, teacher”*.” (STPr-5).

Family education: “*I try to give them information. Whenever we meet, I try to give them something they can tell their children and support them with. I send them weekly newsletters. If they have questions about childcare, I give them relevant information. I explain to them their children’s cognitive, linguistic, social-emotional, and psychomotor development. I do my best to support the parents’ lack of knowledge and skills in these areas*…” (STPr-3).

Tolerance: “*…I try to have a positive approach to both the children and the parents, and I try not to make a big deal out of very small things, and I try to tolerate them. At this point, I also have a lot of meetings with the parents. I try to put a positive spin on things that might happen.*” (PTPr-1).

Empathy: “*When parents see the classroom environment when they see that education is going on and that it’s done here to educate, they understand the difficulty of working. They understand how difficult it is to deal with children. By involving the parents in this way, I try to give them a better understanding, and empathy for our work*…” (STPr-3).

Drawing attention to the rules: “*…I meet with the parents on a one-to-one basis, talk about the rules of the school, explain the situation at the school, and set expectations in this direction to eliminate the stressful situation*.” (PTN-5).

Transfer from experiences: “*Experience is an important factor. Comparing the first years with the current ones, I see that I am calmer. I approach things carefully and when I see a person’s reaction, I am a little more cautious, and a little more explaining. In my opinion, that has a lot to do with experience. With every year that goes by, you definitely put a little bit on top of yourself and you move up like a steppingstone*.” (STPr-2).

Smiling: “…*by smiling (smiles). I don’t have a lot of stress, to be honest with you. I have a relaxed attitude and I am in favor of a relaxed attitude no matter what…*” (PTPr-6).

Staying calm: “*I try to communicate one-on-one with parents in meetings or other gatherings. I also give out my phone number voluntarily. As I respect people, people must respect me… I don’t think I have any wrong behavior, so I am a person who follows my truth by staying calm. People regret their initial behavior when they get to know them over time…*” (STPr-4).

Smoking and taking a break: “*I smoke, take short breaks, and reward myself*.” (PTPr-2).

Ignoring: “*I try to ignore many things. It’s like a kind of desensitization over time*…” (PTPr-5).

There was evidence that using an active behavioral coping strategy was a response to another stressor, the institution/management. The following are examples of the opinions expressed in relation to these codes:

Communication: “*After consulting other teachers and school counselors, we try to cope with stress by using methods such as finding solutions to problems that arise*.” (PTPr-3).

Cooperation: “*We do work with friends in a planned way. We organize the hours we work. We try to avoid getting mixed up. We, for example, try to organize breakfast, dinner, and activity time (for shared areas) efficiently. Problem-solving in co-operation results in less stress for us*.” (PTPr-3).

Professional support: *“…I get professional help…*” (STK-4).

Teachers have been found to use both active behavioral and passive avoidance strategies to manage stressful situations caused by children. Examples of participants’ opinions on these codes are provided below:

Loving the children: “*…constantly doing games, and activities… I hug the children, especially when I have many problems with the 0 to 3-year-olds. Skin contact is so important at this age. I hug, love, kiss, say I love you very much, I’m glad you’re here… This is how we deal with stressful situations*.” (PTN-2).

Spend more time with children: “…*by paying more attention. Spending more time with the children …I don’t want to get away from them at all. By making the time I spend with them more fun, I try to reduce stress*.” (PTPr-4).

Patience and tolerance: “*…I show patience with children who are not toilet trained, who have problems such as not eating, who do not participate in activities, I tolerate their problem behavior. I spend time with them slowly, saying nice words. In this way I try to get them out of negative situations*.” (PTPr-5).

Strict adherence to the rules: “…*it’s more about my behavior and attitudes, so I definitely don’t give up my attitudes. If it’s not going to happen, it’s not going to happen. Not in the form of hard rules, but with precise lines in a language they can understand a little more. I mean, I state it with clear and precise lines so that they understand it. Otherwise, it doesn’t work in any way.*” (PTPr-2).

Trying to be happy: “*…trying to be happy.*..” (STK-1).

Taking precautions: “…*we try to anticipate and prevent incidents from happening. We do our best to minimize negative attitudes on the part of parents through constant communication, and interaction with them within the framework of ethical rules*.” (STPr-6).

Strict control: “*I prefer teacher-centered games. I give tasks to the active students. I try to keep them closed during the activity.*” (STK-6).

### 4.3. Times of Intense Stress

As a result of the interviews with the teachers, the second question asked to the teachers was “At which times of the day do you experience stress while teaching?”. The findings given in Figure 3 emerged from the analysis of the responses to this question.

Preschool teachers reported that they experienced more stress during eating, cleaning, sleeping, arrival and departure times, times when parents communicate, free play times, and times when there are no activities. Examples of opinions in relation to these codes are shown below:

Eating times: “*Mealtimes… Breakfast in the morning is between 8 and 9 in the morning. Lunch is between 12 and 13 in the afternoon. At that time, we were very stressed. We want them to eat, we want them to get up, and we want to do our work, but these times are long, and it is very difficult to keep control of the children*.” (PTN-1).

Cleaning times: “*Hours for handwashing, cleaning, and toileting. In other words, the preparation for the feeding and the feeding time is the hours in which I feel the most stress.*” (STPr-3).

Sleeping times: “*Regarding stress, the mornings come to mind. Because those are the busiest times. In any case, in the afternoon, it is time to sleep, the children do not get up early and rest at noon. There is a difference between the children who get up from their warm beds in the morning and those who rest in the afternoon. It can be more difficult for them than the hours in which they have more energy*.” (PTN-2).

Arrival and departure times: “…*the arrival and departure times of the children at school. Some parents want to drop off their children early, some want to bring them back late and some want to pick them up early. Some parents ask for their child’s hat or scarf when they pick them up from school. Or they scold their child if they can’t find it… Or some parents ask what their child has done during the day, whether he/she has had lunch or not, and open the child’s lunch box to find out. In this situation, it stresses me a lot when they (parents) scold their children when they come to pick them up from school, even though I am so careful with my children*” (STPr-4).

Communication times with the parents: “…*if there is a problem, the parents actually do call or send a text message, but they do it late at night at odd times. I tell them to call at appropriate times, but it doesn’t happen*.” (STPr-6).

Free times to play: “…*playtime and playroom, it is difficult to coordinate children there. Because children are much freer here. So, I am very afraid that something will happen to the children*.” (STPr-6).

Times without activities: “*…I don’t have any difficulties when I am doing the activities myself, but I get more stressed when they are free. I don’t want to give the children too much freedom*.” (STPr-4).

## 5. Discussion

This study explored stressors of preschool teachers, highlighting parents, children, and institutional factors as the main stressors. These stressors were observed both in personal interactions and organizational aspects like school management and leadership style. However, the personnel level (the self-esteem and self-efficacy of the teachers, etc.) is not included in the variables of this study [81]. Thus, this study’s results provide evidence to support the previous assumption that teachers externalize sources of stress [71,72]. The findings also support the confirmation of the contribution to teachers’ stress from different sources such as the physical environment, family, children, and administration [19,82]. Therefore, environmental factors’ impact on stress emerged in this study [16,17,18].

As an important result of this research, all participants identified parents as the first stressor at the interpersonal level. The relationship between the family and the teacher, which is educationally important, can sometimes become an interpersonal stressor that causes teachers to become stressed. This finding has been confirmed by a lot of research [40,83]. Managing parent demands and expectations in teacher–parent communication and interaction for child development is challenging for teachers [37]. In preschool education, teachers and families require a closer relationship. Yet, issues arise due to the perception of preschools as mere child-minding places and teachers as mere caregivers. These problems, frequent in occurrence, rank among the main stressors for teachers. Furthermore, families not adhering to time and place regulations when seeking child-related information also contribute significantly to stress.

Teachers identified children as another significant interpersonal stressor. Challenges stemmed from the young age range (0–3 years old), varying self-care abilities, large class sizes, inclusive student needs, and mixed-age classes. Consequently, teachers encountered difficulties in managing classrooms effectively, limiting free play, enforcing stringent rules, and offering fewer movement-oriented activities. This pattern is consistent with various national and international studies. These studies have delved into factors like challenging child behavior [15,29,41,43,84,85], classroom size [39,40,57], child age [86,87], and the presence of children with special needs [15,88]. All of these variables contribute to the stress experienced by teachers.

However, this study’s most striking finding is that teachers are reluctant to include child-centered play in the program for fear of losing control. In this context, it can be highlighted as an important shortcoming that the program does not include any games that the children plan and over which they have control [68]. Therefore, it can be assumed that the stress experienced by teachers has resulted in teacher-centered activities becoming dominant, play-based preschool education being distracted from its purpose, fun, and active activities that children enjoy being poorly integrated into the daily schedule. Unfortunately, teachers may not be able to implement effective pedagogy and curricula that are appropriate for the development of the children, which are important variables of the quality of early childhood education [89].

Preschool teachers play an important role in nurturing children’s social, emotional, and behavioral development [90]. The quality of teacher–child relationships has a significant impact on young children’s social, emotional, and academic progress. However, increasing workplace stress among teachers can escalate conflict within these relationships [91,92]. Research suggests that escalating behavioral problems in children lead to emotionally negative interactions with teachers, which exacerbate conflicts [93,94]. As a result, deteriorating teacher–child relationships can negatively affect children’s development [95,96]. These negative effects may extend to later stages of education [97].

Preschool teachers view a well-functioning classroom as one that is balanced in terms of gender, age, ethnicity, and class size [87]. However, this perspective can be criticized for a number of reasons. Firstly, the value of peer support in mixed-age groups [98] may have been overlooked. Secondly, inclusive education benefits both special needs and typically developing children [99,100], which may be undermined if teachers perceive special education settings as stressful. Finally, maximizing the benefits of early education is crucial. Viewing younger age groups as stressful and preschool education as suitable only for older children (e.g., 5 years old) contradicts the nature of early education and the scientific evidence [101].

The institutional context (organizational level) contributes significantly to teachers’ stress in preschool education [102]. Recent studies highlight the impact of teaching conditions on stress levels [58]. These include inadequately equipped classrooms [38,39,40], low teacher salaries, limited opportunities for career advancement, interruptions during lessons/activities, heavy workloads, and time constraints [38,103]. Stressors also include negative aspects of school structure and operations [19] and strained working relationships [103]. In addition to other research findings, this study revealed that factors such as inadequate classroom support staff, larger class sizes (ranging from 25 to 30 students), and preparations for national or religious events contribute to teacher stress.

The distinctive demands of preschool education might have contributed to the emergence of these varied stressor types. Given the young age of the children, a classroom assistant is necessary to maintain oversight. Moreover, the teacher must remain vigilant to prevent accidents and conflicts that can arise in a crowded classroom setting. As class size grows, managing the classroom becomes more intricate. Furthermore, facilitating diverse activities that align with the developmental traits of the children presents a challenge.

Administrative policies, another organizational-level stress factor, have an impact on teacher stress [58]. In this regard, research has confirmed that problems in the relationship between school management and teachers [46,83,104,105] affect the stress of early childhood teachers and others [45,106,107]. For example, the manager’s attitude, behavior, management skills, quality of communication processes, and whether or not the manager is supportive are among the stressors for teachers [45]. If teachers feel valued and appreciated and have confidence that the school management has their long-term interests in mind, they are more likely to resist negative influences. Evidence has therefore been obtained for research that expresses the power of the principal in shaping the environment and the structures that enhance or weaken teachers’ work [108]. The findings of this research are a reminder to school administrators of the importance of the role they can play in helping teachers to create a better working climate [29].

In this study, preschool teachers reported using strategies such as effective communication, patience, empathy, and family involvement to manage potentially stressful scenarios with families. The research revealed their use of a coping strategy centered on effective communication and collaboration with the administration—a clear source of stress. In addition, they used coping strategies based on feelings of care, resilience, and tolerance in their interactions with the children who were the ultimate stressors. This finding confirmed that the teachers were using active/active behavioral coping strategies. These strategies can be expressed in terms of rational methods of problem solving (communication, interaction, training, etc.) and the seeking of support (from the supervisor, colleagues, and counseling professionals). In addition to active coping strategies, passive/avoidant coping strategies were used. These strategies include trying to be happy, smiling, staying calm, and ignoring the situation. It was also confirmed that, contrary to these findings, preschool teachers do not use active cognitive strategies to cope with stress [61,73,103,109,110]. The passive strategies used by teachers to cope with stress support the surface-acting dimension of emotional labor [24,25,26]. Teachers were also found not to utilize coping strategies such as regular sleep, exercise, time management activities [14,18,111], and meditation [112].

In addition to sources of stress and coping strategies, it was found that the most stressful times for teachers were when children were eating, cleaning, sleeping, arriving and leaving, and when parents were communicating. During these times, when children are active or have behavioral problems, teachers struggle to maintain control. Furthermore, teachers have meetings and communication with parents and school management during these times. This can be seen as a result of communicating and interacting with the sources that cause stress.

In our study, we found that preschool teachers had negative perceptions of stress and did not view it as an opportunity for personal development. In this regard, recent studies have shown that teachers’ thoughts about stress are good predictors of their psychological well-being as well as their professional development [21]. This is because research shows that teaching-related stress negatively affects the quality of emotional support and classroom organization [113], that emotionally exhausted teachers have lower-quality interactions with children in their classrooms [27], and that this negatively affects many characteristics of children, including their resilience, and even their executive functioning [114,115]. For this reason, professionals working in early childhood, one of the most important periods of children’s development, need to understand how to cope with work stress and, if they cannot, how it affects children’s development.

## 6. Conclusions

In this study, sources of stress for preschool teachers were identified at the interpersonal, and organizational levels. Evidence of the existence of negative effects of stressors externalized by preschool teachers was obtained. Specifically, we found many teacher narratives that led us to conclude that teacher stress is a significant barrier to effective child support. According to the teachers participating in the study, an effective preschool class is characterized by the following features: a small number of children, a small age range of children, the absence of children with behavioral difficulties and children with special needs, and the presence of support staff. From another perspective, the most important implication is that stress may have undermined teachers’ activities supporting children’s development in their classrooms. Children should be provided with play-based experiences in which they can actively participate, discover by doing and experimenting, and have fun to support their early development. However, our research showed that teachers provide fewer opportunities for children to play, to avoid stress. In addition, the teachers have strict control over the children. They expect strict adherence to the rules. They are also challenged to manage behavior problems, create appropriate content and support for children with special needs, manage the classroom, and maintain routines (sleeping, cleaning, eating, etc.). In this context, we have found many variables that negatively affect the quality of early childhood education. In summary, although teachers reported using different strategies (active/behavioral, and passive/avoidant) to cope with stress, we draw attention to the importance of the negative consequences of stress in children. We emphasize that supporting the well-being of teachers and taking steps to reduce the impact of stressors are critical to supporting children’s development in a quality manner, given the unique characteristics of preschools, which are the first institutions where children leave their families.

### Limitations and Recommendations

There are some limitations of our research. The first is that the number of participants was limited. Secondly, the obtained data were devoid of the observations and opinions of male preschool teachers. Finally, the self-esteem and self-efficacy of the teachers, which are personnel-level sources of stress, were not assessed. Therefore, we recommend further research that considers all these limitations. We also believe that the impact of coping resilience intervention programs that support preschool teachers in coping with stress needs to be investigated [46]. For example, we recommend examining the results of experimental studies examining the effects of cognitive behavioral programs [116], meditation or mindfulness-based practices [31,112], and emotion regulation studies [29] on teachers’ stress levels to determine whether they affect child outcomes. Derived from the study’s findings, we present several recommendations deemed pertinent for practical consideration. First and foremost, we advocate for administrators to institute prompt and fitting communication frameworks connecting teachers and parents. Furthermore, we underscore the significance of undertaking initiatives to enhance teachers’ professional progression and overall well-being, concurrently ameliorating their workload. Viable strategies may encompass class size reduction, allocation of support personnel, and the formulation of protocols to enable teachers to effectively manage children during periods of heightened stress. Finally, we propose a study of children’s learning outcomes in the classrooms of teachers with low levels of psychological well-being and low levels of desire for professional development.

## Figures and Tables

**Figure 1 behavsci-14-00059-f001:**
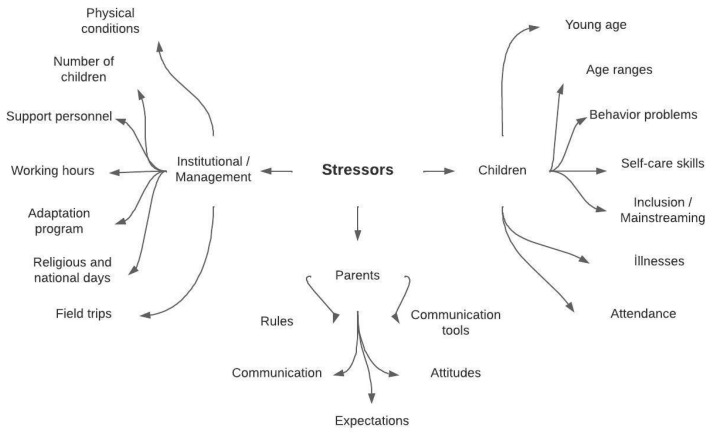
Sources of stress for preschool teachers.

**Figure 2 behavsci-14-00059-f002:**
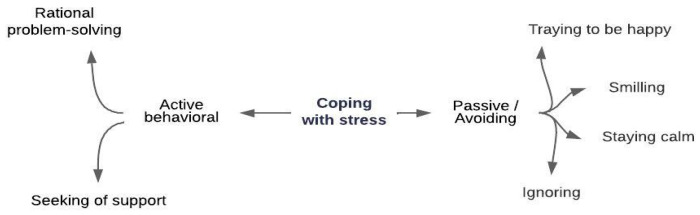
How preschool teachers cope with stress.

**Figure 3 behavsci-14-00059-f003:**
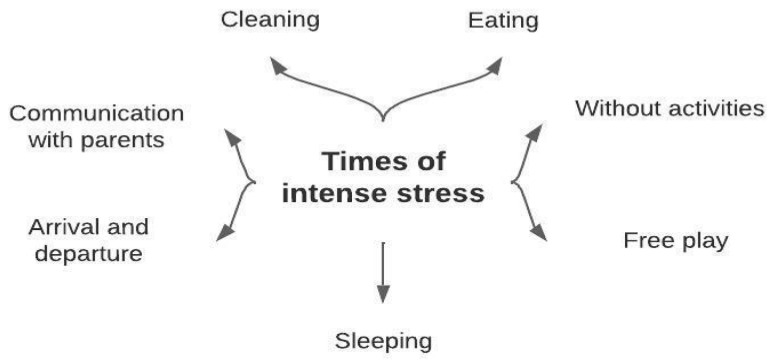
Preschool teachers’ most stressful times.

## Data Availability

The raw data supporting the conclusions of this article will be made available by the authors on request.

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
