# Peer review of "Exploring the Sources of Stress and Coping Strategies of Turkish Preschool Teachers"

_behavsci, 2024, doi:10.3390/bs14010059_

Round 1

Reviewer 1 Report

Comments and Suggestions for Authors

The description is detailed and meets the requirements of qualitative studies. However the questions in the semi-structured interview are over direct and predictable. Rather than asking directly to describe the sources of stress, one could ask indirectly if conditions at work are satisfactory and pleasant or not, and whether examples for conditions can be provided. By asking directly for the sources of stress there is an inherent bias that there is stress, and the interviewee is asked to justify the apriori supposition. 

The findings are rich and intriguing. One example for this is the presentation of communication tools, such as 'whats app groups' as sources of stress. 

Another example for a detailed finding is the time of daily routine and activity as a source of stress within the kindergarten context. 

Though there exists extensive research in the field of workload and teacher stress it is interesting to read a study that focuses on a specific population of preschool teachers in a specific context. 

The literature review is encompassing, yet lacking key terminology in the field, such as: emotional labor, and significant researchers, such as Oplatka, for instance.

The uniqueness of the study as stated by the authors is presenting holistic views of preschool teachers regarding the sources of stress. It is unclear what is meant by holistic. If the views are holistic it would be difficult to juxtapose them with findings in previous studies that are mentioned in the literature review, for instance the finding that preschool teachers do not use cognitive strategies to cope with stress.  

Conclusions

I was looking for an overarching conclusion that would theoretize the findings in light of an educational theory or other. Instead there was a claim that some variables may affect early childhood education negatively. It is unclear how such a conclusion could be derived from the results in this study. 

Comments on the Quality of English Language

minor editing is required

Author Response

Dear Referee:

We express our sincere gratitude for your invaluable contributions to our research. Your feedback has been meticulously reviewed, and we have diligently incorporated the necessary revisions. A comprehensive account of the edits made to our article, along with detailed explanations, has been documented in a separate file, which we are pleased to present for your perusal.

We respectfully draw your attention to the appended document.

With the utmost respect

Reviewer 2 Report

Comments and Suggestions for Authors

The article reports on research with pre-school teachers and explores their views on the stressors in their work practice, their frequency and the ways in which they cope with them.

With regard to the introduction, the authors provide context about work-related stress, focusing on stress in teachers. The introduction also introduces the potential reader to the preschool system and preschool teacher education in Turkey, which contributes to their readability. However, authors are advised to consider the most recent and international academic literature in the Introduction section, as only a small percentage of the references are from the last 5 years, and the topic has been studied extensively by their own admission.

Both research design and analysis are adequate. It is suggested to justify the reason for the feminisation of the sample on the basis of the Turkish case: number of male/female teachers in pre-school education, number of women and men working in kindergartens, etc.

The authors include the following paragraph at the end of the Discussion section: “We hypothesise that teachers with low levels of psychological well-being, and a lack of desire for professional development will not do well in their classrooms.” The authors are advised to dispense with a hypothesis that has not been investigated in their study. It could be proposed as a future line of study in which the possible correlation between levels of psychological well-being and teachers' classroom performance will be investigated.

On the other hand, authors are encouraged to include limitations and recommendations in the Discussion section rather than in the Conclusions section.

Comments on the Quality of English Language

No comments on the quality of English Language.

Author Response

(The authors gave the same response as above.)
